# Complications and Discharge after Radical Cystectomy for Older Patients with Muscle-Invasive Bladder Cancer: The ELCAPA-27 Cohort Study

**DOI:** 10.3390/cancers13236010

**Published:** 2021-11-29

**Authors:** Romain Geiss, Lucrezia Sebaste, Rémi Valter, Johanne Poisson, Soraya Mebarki, Catherine Conti, Dimitri Vordos, Michael Bringuier, Arnaud Méjean, Pierre Mongiat-Artus, Tristan Cudennec, Florence Canoui-Poitrine, Philippe Caillet, Elena Paillaud

**Affiliations:** 1Department of Geriatrics, AP-HP, Hôpital Européen Georges Pompidou, 75015 Paris, France; romain.geiss@aphp.fr (R.G.); lucreziasebaste@hotmail.it (L.S.); johanne.poisson@aphp.fr (J.P.); soraya.mebarki@aphp.fr (S.M.); catherine.conti@aphp.fr (C.C.); philippe.caillet@aphp.fr (P.C.); 2Department of Public Health, AP-HP, Hôpital Henri-Mondor, 94000 Créteil, France; remi.valter@aphp.fr (R.V.); florence.canoui-poitrine@aphp.fr (F.C.-P.); 3Center for Research on Inflammation, Université de Paris, Inserm U1149, 75018 Paris, France; 4Department of Urology, AP-HP, Hôpital Henri-Mondor, 94000 Créteil, France; dimitri.vordos@hmn.aphp.fr; 5Department of Supportive Care and Medical Oncology, Institut Curie, 92210 Saint-Cloud, France; michael.bringuier@curie.fr; 6Department of Urology, AP-HP, Hôpital Européen Georges Pompidou, 75015 Paris, France; arnaud.mejean@aphp.fr; 7Department of Urology, AP-HP, Hôpital Saint Louis, 75010 Paris, France; pierre.mongiat-artus@aphp.fr; 8Department of Geriatrics, AP-HP, Hôpital Ambroise-Paré, 92100 Boulogne-Billancourt, France; tristan.cudennec@aphp.fr; 9Université Paris-Est, Inserm, IMRB, 94000 Créteil, France; 10Faculty of Health, Université de Paris, 75006 Paris, France

**Keywords:** radical cystectomy, geriatric assessment, older patient, complications, length of hospital stay, functional decline, discharge

## Abstract

**Simple Summary:**

Radical cystectomy is associated in older patients with an increased risk of post-operative complications. However, these studies did not take into account geriatric variables. In our study, all patients had a standardized geriatric assessment prior to radical cystectomy. Although geriatric variables were not associated with 30-day complications, our study found that frailty (measured as a G8 score ≤ 14), a loss of autonomy, anemia, and severe comorbidities were associated with a higher risk of not being discharged home at one month after the surgery.

**Abstract:**

Radical cystectomy is the standard of care for localized bladder cancer but is associated with high morbidity and mortality rates—especially among older patients with comorbidities. The association between geriatric assessment parameters on post-operative complications and discharge has not previously been investigated. The present analysis of the Elderly Cancer Patient (ELCAPA) prospective cohort included all patients aged ≥70 having undergone a geriatric assessment and then radical cystectomy for localized muscle-invasive bladder cancer between 2007 and 2018. The primary endpoint was the proportion of patients with one or more complications in the first 30 days after cystectomy. The secondary endpoints were the length of hospital stay (LOS), the 30-day mortality, and discharge rates. Sixty-two patients (median age: 81; range: 79–83.8) were included. The 30-day complication rate was 73%, and 49% of the patients had experienced a major complication, according to the Clavien-Dindo classification. The 30-day mortality rate was 4%. None of the geriatric, oncological, or laboratory parameters were significantly associated with the occurrence or severity of complications. The median (interquartile range) LOS was 18 days (15–23) overall and was longer in patients with complications (19 days vs. 15 days in those without complications; *p* = 0.013). Thirty days after cystectomy, 25 patients (53%) had been discharged to home and 22 (47%) were still in a rehabilitation unit. In a univariate analysis, a Geriatric-8 score ≤ 14, a loss of one point on the Activities of Daily Living Scale, anemia, at least one grade ≥ 3 comorbidity on the Cumulative Illness Rating Scale-Geriatric, and an inpatient geriatric assessment were associated with a risk of not being discharged to home. In older patients having undergone a geriatric assessment, radical cystectomy is associated with a high complication rate, a longer LOS, and functional decline at 30 days.

## 1. Introduction

Bladder cancer is a disease of older people; the median age at diagnosis is 73 [1]. Radical cystectomy (RC) with or without perioperative chemotherapy constitutes the standard of care for localized, muscle-invasive bladder cancer [2]. Even though the frequency of RC has increased over recent years, patients aged 80 or over are less likely to receive this treatment than their younger counterparts (20% vs. 55%, respectively) [3]. Indeed, RC is a major surgical procedure with high morbidity and mortality rates, which increase with age [4]. For example, mortality rates as high as 10% have been reported for patients in their 80s [5].

Considering that smoking is the main risk factor for bladder cancer, patients often have smoking-related cardiovascular and respiratory comorbidities. The median number of chronic comorbidities is eight for patients with bladder cancer, compared with four in non-cancer patients [6]. The comorbidity burden (evaluated with the Charlson Comorbidity Index or the American Society of Anesthesiology (ASA) score, for example) is also associated with higher morbidity and mortality rates following RC [7]. Furthermore, several studies have shown that frailty (measured with various tools, such as the modified frailty index (mFI)), is an independent risk factor for adverse events following RC [8].

A geriatric assessment (GA) is a multidimensional evaluation of different domains: functional status (including the fall risk), nutrition, cognitive status, mood, comorbidities, polypharmacy, sensory impairments, and social support. A GA is recommended by the International Society of Geriatric Oncology [9] and the American Society of Clinical Oncology [10] for patients at risk of frailty (as identified with screening tools like the Geriatric-8 (G8) and the Vulnerable Elders Survey (VES-13)) prior to deciding on the cancer treatment. Although the GA’s domain scores predict postoperative outcomes in oncology in general, specific data for RC are lacking.

The objectives of the present study were to assess the 30-day complication rate; the 30-day mortality rate; the length of hospital stay (LOS); the 30-day discharge rate; and the prognostic value of the GA scores, oncologic variables, and routine blood test results.

## 2. Materials and Methods

### 2.1. Study Design

ELCAPA is a prospective, multicenter study of a cohort of patients aged 70 and over with a solid tumor cancer and who have been referred to a geriatric oncology clinic for a GA prior to selection of a cancer treatment in 19 hospitals in the greater Paris region of France. The ELCAPA cohort’s GA has been described previously [11]. All patients provided their written informed consent prior to inclusion. The ELCAPA study protocol has been approved by an institutional review board (CPP Ile-de-France I, Paris, France; reference: 2019 mai-MS121). ELCAPA cohort is registered on ClinicalTrials.gov (NCT02884375).

In the present analysis (ELCAPA-27), we included all ELCAPA patients with a localized muscle-invasive bladder cancer treated with RC between 2007 and 2018 at seven of the 19 hospitals (Henri-Mondor, Hôpital Europeéen Georges Pompidou, Cochin, Saint-Louis, Paul-Brousse, Bretonneau, and Institut Curie).

The non-inclusion criteria were metastatic cancer, no RC, non-invasive bladder cancer, a second cancer (i.e., in addition to bladder cancer), and the lack of data on postoperative follow-up (Figure 1).

### 2.2. Data Collection

We prospectively collected data on the patients’ general and geriatric characteristics from the GA performed before surgery: age, sex, smoking history, functional status (performance status, a six-item activities of daily living (ADL) score, and an eight-item instrumental activities of daily living (IADL), mobility (falls in the past 6 months, a timed up-and-go (TUG) test, and the one-leg standing balance test), nutritional status (weight loss ≥5% in the previous month and/or ≥10% in the previous 6 months, body mass index, the Mini Nutritional Assessment score), cognitive status (the Mini-Mental State Examination), mood (the four-item Mini-Geriatric Depression Scale), comorbidities (on the Cumulative Illness Rating Scale-Geriatric (CIRS-G)), polypharmacy (four or more prescription medications), sensory disorders (eyesight and hearing), and the social environment (living alone or not).

The prospectively collected pre-operative laboratory variables were the blood hemoglobin level, the white blood cell count, and the serum creatinine, calcium, and albumin levels. Anemia was defined as a hemoglobin level below 13 g/dL for adult males and 12 g/dL for adult females, neutrophilic leukocytosis was defined as a neutrophil count above 7.5 G/L, and hypercalcemia was defined as a corrected serum calcium level above 2.6 mmol/L.

The TNM classification was based on the surgical pathology report. Data on neoadjuvant chemotherapy and radiotherapy were collected prospectively. The ASA score and data on the type of urinary diversion were collected retrospectively.

### 2.3. Outcomes

The primary endpoint was the 30-day post-operative complication rate. Complications were defined as surgical complications (bleeding, evisceration, surgical site infection, occlusion, or fistula), non-surgical complications (non-surgical site infection, thrombo-embolic complications, cardiovascular complications, delirium, falls, bed sores, and other complications), and death. Complications were evaluated according to the Clavien-Dindo classification: grade I and II complications were considered to be minor, and grade III, IV, and V complications were considered to be major. When a given patient had more than one complication, the most severe Clavien-Dindo grade was considered in the analysis.

The secondary endpoints were the LOS and the 30-day mortality and discharge rates.

### 2.4. Statistical Analysis

The study populations’ characteristics were described using summary statistics. We made no assumptions about missing data and expressed proportions with regard to the number of patients with data. Categorical variables were described as the frequency (percentage), and continuous variables were described as the mean (standard deviation) or the median (interquartile range), depending on the data distribution. Intergroup differences were assessed using the Wilcoxon rank sum test or the Kruskal–Wallis test for continuous variables and the chi-squared test or Fisher’s exact test for categorical variables. Kaplan-Meier survival curves were plotted using the survival package in R. All tests were two-sided and a *p* value < 0.05 was considered significant. All statistical analyses were performed using R software v. 4.0.2 R Foundation for Statistical Computing, Vienna, Austria.

## 3. Results

### 3.1. Population Characteristics

A total of 62 patients having undergone RC were included in the study (Table 1). The median age was 81 (range: 79–83.8). The majority of the patients were male (77%) and current or former smokers (72%). The tumors were locally advanced: 48% were graded as ≥pT3 and 54% had positive lymph nodes.

The G8 score was ≤14 for 69% of the patients. The pre-surgical functional status was generally maintained after RC, with a PS of 0 or 1 in 81% of the patients, an ADL score of 6 out of 6 in 87%, and an IADL score of 8 out of 8 in 52%. Mobility was normal (timed up and go test result ≤20 s) for 89% of the patients, and 21% of the patient had a history of falls in the previous six months. Furthermore, 72% and 85% of the patients had not lost weight in the previous six months or the previous month, respectively. The median serum albumin level was 35 g/L. The median CIRS-G score was 11, and 54% of the patients had one or more grade 3 or 4 comorbidities. Polypharmacy was observed in 64% of the patients.

### 3.2. Thirty-Day Complication and Mortality Rates

In the 30 days following RC, 45 patients (73%) presented at least one complication and 17 (27%) did not present any complications. The total number of complications was 100 (69 non-surgical complications and 31 surgical complications; Table 2). According to the Clavien-Dindo classification, grades I and II (minor) complications were observed in 23 patients (51%), and grade III, IV and V (major) complications were observed for 22 (49%) patients, 3 of whom died. The most frequent grade was grade II (Figure 2). The 30-day mortality rate was 4%.

None of the geriatric, oncologic, or laboratory variables was significative associated with the occurrence or severity of complications (Appendix A).

### 3.3. LOS and Discharge

Overall, the median LOS was 18 days (interquartile range (IQR): 15–23). The LOS was significantly longer (*p* = 0.013) for patients with complications than for patients without complications, with median values of 19 days (IQR: 15–25) and 15 days (IQR: 13–16), respectively (Appendix A).

Data on 30-day discharge were available for 47 patients: 25 patients (53%) had been discharged to home, and 22 (47%) were still in a rehabilitation unit. In a univariate analysis, a G8 score ≤ 14, an ADL score < 6, anemia, at least one CIRS-G grade 3 or 4 comorbidity, and inpatient status at the time of the pre-operative GA were significantly associated with being in a rehabilitation unit 30 days after surgery (Table 3).

## 4. Discussion

In the present study, RC after a GA was associated with a high 30-day complication rate (73%). Half of these complications were major, according to the Clavien-Dindo classification. Patients with complications had a longer LOS, and almost half the patients were still in a rehabilitation unit 30 days after surgery. None of the oncologic, GA-related, or laboratory parameters were associated with complications.

The 30-day complication rate of 73% was slightly higher than that recorded for patients of all ages after RC. However, the proportion of our patients with major complications was markedly higher than in the literature. In the large MSKCC prospective cohort (*n* = 1142; median age: 68), the 30-day complication rate was 58% overall but only 18% when considering major complications [12]. Our data are also consistent with Berneking et al.’s reported value of 86% overall and 33% for major complications (according to the Clavien-Dindo classification) in a group of 43 octogenarians [13].

Advanced age (especially after 80) is associated with an elevated risk (by up a factor of 5 vs. younger patients) of postoperative death [4]. In our study, the mortality rate was twice as high as in all-age population studies (2–3%). However, our study population was strongly selected; the GA excluded very frail patients for whom RC was not indicated.

We did not find any geriatric, oncologic, or laboratory variables associated with morbidity and mortality; this was probably due to the small number of patients in our study and thus a lack of statistical power. A recent study by Chesnut et al. found that patients aged 75 and over with impaired domains in a pre-operative electronic rapid fitness assessment (timed up and go test, cognitive function, social activity, distress, and polypharmacy) were more likely to (i) attend the emergency department, (ii) be admitted to an intensive care unit, and (iii) experience major (Clavien-Dindo grade III–V) complications after RC [14].

In our study, the median LOS stay was 18 days overall and was longer in patients with complications (19 days vs. 15 days in patients without complications). This LOS is twice as long as in the general population (8–11 days) [15] and emphasizes the association between frailty and a longer LOS [16]. Comorbidities, a low albumin level, male sex, and older age are also known to increase the LOS after cancer surgery [17,18,19]. In our study, all the patients were treated in high-volume centers; according to the literature data, a high number of urologic cancer operations is associated with a lower risk of complications, a shorter LOS, and a lower mortality rate [2].

Half of our study participants were still in a rehabilitation unit 30 days after RC. This rate is very high, given that the value in an all-age population of patients discharged to a care facility or a rehabilitation unit after RC is around 15% [20]. This difference might reflect a major, prolonged loss of autonomy after RC in our geriatric population. Indeed, Murray et al. showed that patients who needed a care facility after RC had impaired ADL [21].

Factors associated with maintenance in a rehabilitation unit 30 days after RC were a G8 ≤ 14, loss of autonomy for ADL, anemia, severe comorbidities on the CIRS-G, and an inpatient GA. Chesnut et al. found that patients with PS, ADL, IADL, or TUG impairments, falls in the past year, low social activity, weight loss, and self-reported depression were less likely to be discharged to home after RC [14]. Pearl et al. showed that frail patients (according to the mFI) had a 2.33-fold greater risk of non-home discharge, and that the latter risk was even higher for patients with a major in-hospital complication [16].

Osterman et al. performed the only study to have included a GA before RC and at various times during 12 months of follow-up [22]. The researchers evaluated recovery in 80 patients aged 70 and over, in comparison to younger patients. An impairment in physical function was defined by an impairment in at least one of the following outcomes: PS, the IADL score, the TUG test, falls, or the Medical Outcomes Study Physical Health survey. At one month, both age groups had showed a worsening in physical function, but the decline was greater in the over-70 group: more than 90% of the patients had at least one impairment in physical function. At 3 months, both groups had returned to baseline levels of physical health, quality of life, and social activities. In Osterman et al.’s study, 5% of the patients were discharged to a care facility.

With a view to reducing post-RC complications, guidelines on enhanced recovery programs have been published [23]. It has been reported that multidisciplinary geriatric patient management by surgeons, anesthesiologists, and geriatricians with direct control over medical issues during the post-operative period is associated with a significant reduction in 90-day postoperative mortality among older patients with cancer [24].

The present study had some limitations. Firstly, there is a selection bias due to the low incidence of patients included and receiving pre-operative GA. One explanation is that the ELCAPA cohort did not open at the same time in all the hospitals (initially monocentric cohort) and GA was not at first able to assess patients in all tumor boards. Another explanation is that frailty, using G8 or other tools, was not systematically used by urologists. Additionally, our cohort did not capture fit patients receiving surgery with no GA or those declined or refusing surgery or receiving radiation therapy. Therefore, our results cannot be extrapolated to the population of older patients undergoing RC in general. Secondly, data on post-operative complications were collected retrospectively from hospital reports, and so some complications might have been missed—especially delirium or falls that are specific to geriatric patients. However, the main strength of our study was the administration of a standardized, prospective GA to all our patients in high-volume cancer surgery centers.

## 5. Conclusions

Older patients selected for RC after a GA had a high 30-day complication rate. Over half the patients were still in a rehabilitation unit 30 days after RC.

Early, long-term, multidisciplinary patient management might help to slow functional decline.

Before RC, older adults should be informed of the surgery’s impact on functional status and likelihood of prolonged hospitalization, especially frail patients. Further research is needed to determine which older patients are at risk of a prolonged loss of functional status after RC.

## Figures and Tables

**Figure 1 cancers-13-06010-f001:**
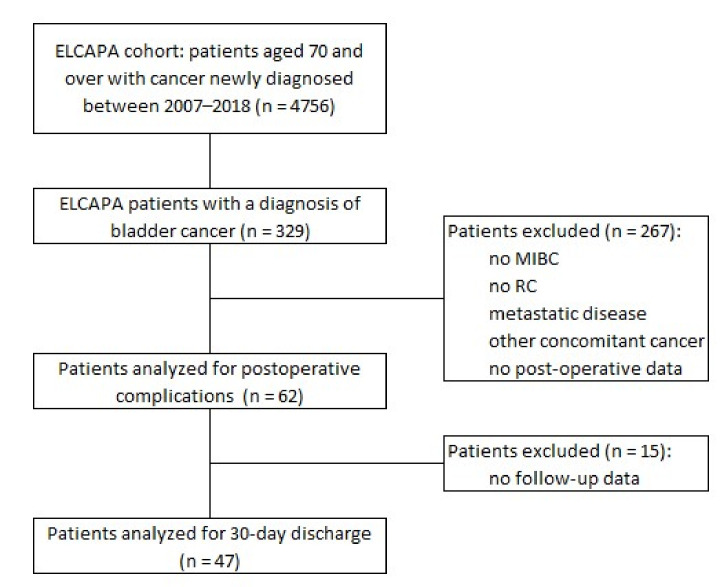
Flow diagram of participants. MIBC: muscle-invasive bladder cancer; RC: radical cystectomy.

**Figure 2 cancers-13-06010-f002:**
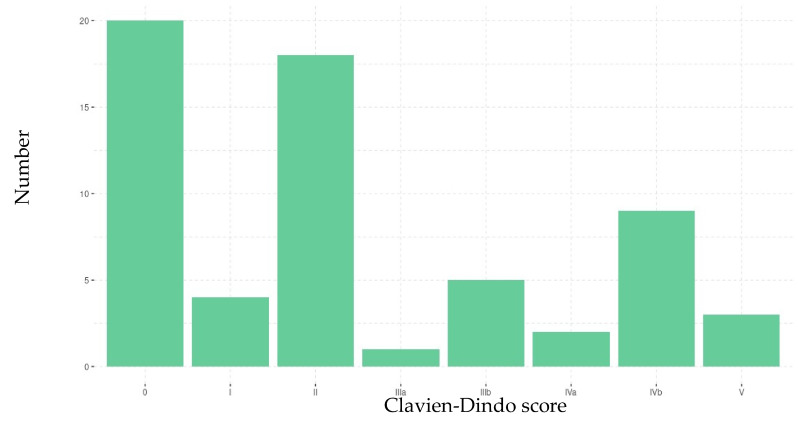
Severity of the patients’ complications, according to the Clavien-Dindo classification.

**Table 1 cancers-13-06010-t001:** Characteristics of the study population.

Patients Characteristics	*n* (%)
Total	62 (100)
Age (years)	
Median (IQR)	81.0 (79.0–83.8)
Sex	
Male	48 (77.4)
Consultation	
Inpatient	8 (13)
Outpatient	54 (87)
G8	
Median (Min, Max)	12.5 (4.00,16.0)
≤14	43 (69.4)
>14	10 (16.1)
Missing	9 (14.5)
Social environment	
Living alone	16 (26)
Functional status	
ECOG-PS	
0–1	50 (80.6)
2	10 (16.1)
3–4	2 (3.3)
ADL score	
≤5/6	8 (13)
IADL score	
≤7/8	30 (48.4)
Tobacco smoking	
Current smoker	6 (9.7)
Former smoker	37 (59.7)
Never-smoker	17 (27.4)
Missing	2 (3.2)
Comorbidity	
Median CIRS-G (IQR)	12 (8–14)
≥1 grade 3–4, CIRS-G	31 (54)
Polypharmacy	
>4 drugs per day	38 (61.3)
Cognitive impairment	
MMSE score <24	6 (9.7)
Depressive disorder	
Mini-GDS score ≥1	15 (24.2)
Mobility	
Falls in the past 6 months	13 (21)
Timed up and go test ≤20 s	51 (82)
One-leg standing balance test <5 s	27 (43.5)
Sensory impairment	
Eyesight	9 (14.5)
Hearing	29 (46.8)
Nutritional status	
Weight loss (%)	
in the last 1 month	9 (14.5)
in the last 6 months	17 (27.4)
Albuminemia (g/L)	
Median (IQR)	35 (24–38)
BMI (kg.m^2^)	
<21	12 (19.3)
21–25	17 (27.4)
25–30	25 (40.3)
>30	8 (13)
Carcinologic and surgical characteristics	*n* (%)
pTNM	
T	
0	4 (7.14)
1	9 (16.1)
2	8 (14.3)
3	20 (35.7)
4	10 (17.9)
In situ associated	5 (8.93)
Missing data	6
N	
0	37 (66.1)
1	6 (10.7)
2	6 (10.7)
3	3 (5.36)
4	0 (0)
x	4 (7.14)
Missing data	6
Neoadjuvant chemotherapy	3 (4.8)
Preoperative radiotherapy	1 (1.6)
Urinary diversion	
Bricker (ileal conduit)	60 (96.8)
Bilateral cutaneous ureterostomy	1 (1.6)
Neobladder	1 (1.6)
ASA score	
Median (IQR)	3 (2–3)
Resection margins	
R0	41 (82)
R1	8 (16)
R2	1 (2)
Missing data	6
Laboratory variables	*n* (%)
Serum creatinine (µmol/L)	
Median (IQR)	98 (77–120)
Missing	1 (1.6)
Hemoglobin (g/dL)	
Median (IQR)	12 (11–13)
Missing data	1 (1.6)
Anemia	
Yes	32 (51.6)
No	29 (46.8)
Missing data	1 (1.6)
White blood cell count (mm^3^)	
Median (IQR)	7900 (6200–8900)
Missing	2 (3.2)
Neutrophilic leukocytosis	
Yes	13 (21)
No	46 (74.2)
Missing data	3 (4.8)
Hypercalcemia	
Yes	0 (0)
No	36 (58)
Missing data	26 (42)

ECOG/PS: Eastern Cooperative Oncology Group/Performance Status; ADL/IADL: activities of daily living/instrumental activities of daily living; CIRS-G: Cumulative Illness Rating Scale-Geriatric; MMSE: Mini Mental State Examination; GDS: Geriatric Depression Scale; BMI: body mass index; ASA: American Society of Anesthesiologists.

**Table 2 cancers-13-06010-t002:** Post-operative complications.

Post-Operative Complications	*n* (%)
Surgical complications	31 (100)
Surgical site infection	8 (25.8)
Bleeding	0 (0)
Evisceration	7 (22.6)
Occlusion	11 (35.5)
Urinary fistula	1 (3.2)
Digestive fistula	4 (12.9)
	*n* (%)
Non-surgical complications	69 (100)
Non-surgical site infection	23 (33.3)
Thromboembolic complication	6 (8.7)
Cardiovascular complication	17 (24.6)
Delirium	8 (11.6)
Pressure sores	1 (1.5)
Fall	2 (2.9)
Others	12 (17.4)

**Table 3 cancers-13-06010-t003:** Univariate analyses of factors associated with the 30-day outcome (discharge or not).

30-Day Outcome	Discharged to Home *n* (%)	Presence in a Rehabilitation Unit *n* (%)	*p* Value *
Total	25 (53.2)	22 (46.8)	
Median age (years) [IQR]	81 (76–83)	83 (80–84)	0.07
Sex			0.3
Male	21 (84)	15 (68)	
Tobacco smoking			1
Current smoker	2 (8)	2 (9.1)	
Former smoker	15 (60)	12 (54.5)	
Never-smoker	8 (32)	6 (27.3)	
Missing	0 (0)	2 (9.1)	
G8 ≤ 14/17	14 (64)	17 (94)	0.03
Consultation			0.04
Inpatient	1 (4)	6 (27)	
Outpatient	24 (96)	16 (73)	
Social environment			1
Living alone	7 (28)	7 (32)	
Functional status			
ECOG-PS			0.31
0–1	20 (80)	15 (68.2)	
2	5 (20)	5 (22.7)	
3–4	0 (0)	2 (9.1)	
ADL score ≤ 5/6	1 (4)	7 (32)	0.02
IADL score ≤ 7/8	12 (48)	11 (50)	1
Mobility			
Falls in the previous 6 months	3 (12)	7 (32)	0.15
Timed up and go test ≤ 20 s	22 (88)	15 (68.2)	0.64
Nutritional status			
Weight loss in the last 1 month	3 (12)	4 (18.2)	0.69
in the last 6 months	4 (16)	6 (27.3)	0.47
Median albuminemia (g/L) (IQR)	37 (24–38)	34 (22–42)	0.51
BMI (kg.m2)			0.36
<21	5 (20)	6 (27.3)	
21–25	5 (20)	6 (27.3)	
25–30	14 (56)	7 (32)	
>30	1 (4)	3 (13.5)	
Cognitive impairment (MMSE score < 24/30)	2 (8)	4 (18.2)	0.4
Missing	4 (16)	3 (13.6)	
Depressive disorder (Mini-GDS score ≥ 1/4)	3 (12)	6 (27.3)	0.26
Missing	3 (12)	3 (13.6)	
Comorbidity, ≥1 grade 3–4, CIRS-G	9 (36)	17 (77.3)	0.007
Missing	2 (8)	1 (4.5)	
Polypharmacy (>4 drugs per day)	18 (72)	14 (63.6)	0.34
Sensory impairment			
Eyesight	4 (16)	2 (9.1)	0.67
Hearing	10 (40)	11 (50)	0.56
Anemia	8 (32)	15 (68)	0.02
Hypercalcemia	0 (0)	0 (0)	1
Neutrophilic leukocytosis	4 (16)	5 (22.7)	0.71
Missing	1 (4)	1 (4.6)	
Median LOS (IQR)	17 (15–20)	21 (15–33)	0.09

ECOG/PS: Eastern Cooperative Oncology Group/Performance Status; ADL/IADL: Activities of Daily Living/Instrumental Activities of Daily Living; CIRS-G: Cumulative Illness Rating Scale-Geriatric; MMSE: Mini Mental State Examination; GDS: Geriatric Depression Scale; BMI: body mass index; LOS: length of hospital stay. * Fisher’s exact test for categorical variables and Wilcoxon’s rank sum tests for continuous variables.

## Data Availability

Restrictions apply to the availability of these data. Data was obtained from the ELCAPA Study Group and are available from the corresponding author with the permission of the ELCAPA Study Group investigators.

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
