# Peer review of "Complications and Discharge after Radical Cystectomy for Older Patients with Muscle-Invasive Bladder Cancer: The ELCAPA-27 Cohort Study"

_cancers, 2021, doi:10.3390/cancers13236010_

Round 1

Reviewer 1 Report

This study demonstrates the relationship between radical cystectomy and geriatric variables. In clinical practice, it is often difficult to decide whether to perform radical cystectomy in the elderly, and this paper seems to be very useful in this regard. Further improvements are expected by clarifying the following points.

1) Since postoperative complications are examined, please include the data of the intraoperative factor which greatly contributes to it. Examples are surgical procedures (open or minimally invasive surgery), intraoperative complications, and the amount of blood lost during surgery.

2) Although a significant percentage of patients have lymph node metastases, radical surgery may be difficult without preoperative chemotherapy in patients with suspected lymph node metastases. Why is surgery performed in such cases? Is it for local control rather than for radical purposes?

3) I think the rate of major complications is quite high. What kind of complications are common? Please describe the details of grade 3-5 complications if possible.

4) What proportion of muscle-invasive bladder cancers without metastasis ultimately chose radical cystectomy in this cohort?

Author Response

1) As you mentionned, robot-assisted radical cystectomy is indeed associated with lower severe post operative complications and blood loss than open radical cystectomy as reported in a recent metaanalyses of Naichu Zou et al. Unfortunately our data do not allow us to collect this per operative variables. Only the type of derivation was collected (97% were Bricker) with no impact on complications.

This limit will be added in the discussion if our manuscript is accepted.

2)Since polychimiotherapy (MVAC) or bichimiotherapy associated with Cisplatin are the standard regimen for neoadjuvant treatment in bladder cancer,  surgery is often not proposed frontline for frail patients with cardiovascular comorbidities or renal impairment which make them ineligible for these protocoles. All patients had radical carcinologic aim in our cohort.

3) The details of grade 3-5 are presented  in thoses 2 tables (attached document) and will be added in the annexe if our manuscript is accepted.

4) Only patients receiving radical cystectomy were included in our study. The proportion of patients with localized bladder cancer refusing or non eligible for surgery were not collected or analyzed.

Reviewer 2 Report

The authors report on a prospective trial evaluating 30-day complöication and mortality rates after radical cystectomy in elderly patietns. The strength of the study is the association of geriatric assessment with the defined study endpoints.

Comments:

30-day time frame alone is relatively short in this population. As recently published in a systematic review, both complication and mortality rates show a substantial increase between days 30 and 90 after cystectomy (Maibom et al, BMJ open 2021). Thus, inclusion of 90-day complication and mortality rates would be of high relevance.

Against the background of a study period of almost 12 years and seven major hospitals involved, the sample size is disappointingly low. At average, less than one patient per hospital and year was recruited. This deserves some discussion. Were perhaps the patients unwilling to undergo geriatric assessment because of the emotional burden of cancer? Any other explanation? The true  incidence of MIBC in patients >70 must be much higher.

Consequently, the study is very likely underpowered and unfortunately results in a lack of statistically significant associations between the assessed parameters and complications. However, a continuation of the study to increase statistical power is highly encouraged, since the issue has high clinical relevance. The intention should be to develop a kind of score and a cut-off, which defines a group of patients for whom the risk-benefit-ratio of cystectomy is unfavorable and thus should be better switched towards radiotherapy.

Reference#1 is not related to the topic.

Author Response

1) We agree with reviewer that 90 day complications and mortality are higher and is more relevent in older and frail patients. Unfortunately since morbidity was collected retrospectively in our study we choose not to assess 90 days morbidity to avoid underestimation of collected data. For information mortality was higher (6%) at 90 days in our cohort and can be added in our results if needed.

2) Dear reviewer, we are also very disapointed with this low incidence for patient receiving CGA. One explaination is that the ELCAPA cohort did not open at the same time in all the hospitals (initialy cohort monocentric) and CGA was not at first able to assess patients in all tumor boards (GI initially). Also, our cohort did not capture fit patients receiving surgery with no CGA our those declined or refusing  surgery or receiving radiation therapy. Most of our patients came from 3 of the 7 hospitals. This small number of patients receiving CGA before this major surgery in our cohort emphasis the work remaining to raise awareness for screening frailty in urological surgeons.

This limitation and sesection biais will be more developed in our discussion if our manuscript is accepted.

3) reference 1 will be changed

Round 2

Reviewer 2 Report

The authors have responded and implemented the suggestions as much as possible. Despite some limitations, I would recommend publication due to the prospective design and novelty of the concept.